# Chemical Screening of Nuclear Receptor Modulators

**DOI:** 10.3390/ijms21155512

**Published:** 2020-07-31

**Authors:** Mari Ishigami-Yuasa, Hiroyuki Kagechika

**Affiliations:** Institute of Biomaterials and Bioengineering, Tokyo Medical and Dental University (TMDU), 2-3-10 Kanda-Surugadai, Chiyoda-ku, Tokyo 101-0062, Japan; marichem@tmd.ac.jp

**Keywords:** chemical screening, nuclear receptor modulators, high-throughput screening, chemical library

## Abstract

Nuclear receptors are ligand-inducible transcriptional factors that control multiple biological phenomena, including proliferation, differentiation, reproduction, metabolism, and the maintenance of homeostasis. Members of the nuclear receptor superfamily have marked structural and functional similarities, and their domain functionalities and regulatory mechanisms have been well studied. Various modulators of nuclear receptors, including agonists and antagonists, have been developed as tools for elucidating nuclear receptor functions and also as drug candidates or lead compounds. Many assay systems are currently available to evaluate the modulation of nuclear receptor functions, and are useful as screening tools in the discovery and development of new modulators. In this review, we cover the chemical screening methods for nuclear receptor modulators, focusing on assay methods and chemical libraries for screening. We include some recent examples of the discovery of nuclear receptor modulators.

## 1. Introduction

Recent progress in chemical analysis, separation, and identification technologies, as well as in the biomedical sciences, has dramatically improved the effectiveness and scope of chemical screening. Schreiber introduced the concept of “chemical biology”, i.e., using small molecules and chemical libraries to systematically explore biology and for drug discovery [1], and this idea has shaped research in this field to a great extent. The development of combinatorial chemistry to generate large chemical libraries has made it possible for biologists to search very large chemical spaces [2,3,4]. Virtual screening is also a useful method to find novel compounds without the need for time-consuming and expensive synthetic efforts [5]. Besides large-scale libraries built by pharmaceutical companies for their in-house use, increasing numbers of commercially available compound libraries generated by academic institutions and venture companies have become available [6,7]. Moreover, public institutions have begun distributing their chemical libraries [8]. Thus, there is an increasing need for efficient evaluation methods to screen very large numbers of compounds. By the 1990s, improvements in the cost per assay and methodology enabled the simultaneous evaluation of more than 1500 samples, which led to the coining of the term “high-throughput screening (HTS)” [9]. There is no doubt that HTS has contributed to the discovery of many biologically active molecules and also encouraged the development of many peripheral technologies, such as automation and data processing [10]. This has expanded the range of assays amenable to the HTS format. Robust and inexpensive assay methods, such as enzyme-based colorimetric assays can also be used in this context [11]. Furthermore, phenotypic and genotypic assays are available to study a broad range of parameters, including morphological change, differentiation, and cell viability, although false-positive results can be an issue [12,13]. Recently, many screening methods targeting molecular interactions, such as small molecule–protein and protein–protein interactions have been developed, especially based on technologies using fluorescent molecules [14,15,16].

Thus, progress in chemical screening has resulted in the huge expansion of available libraries and the development of sophisticated equipment and assay methods that can facilitate not only drug discovery, but also the screening of modulators of a range of biological functions. In this review, we focus on nuclear receptors as target biosubstances. Nuclear receptors are ligand-inducible transcription factors [17]. There are several types of modulators of nuclear receptors, including the specific ligands (agonists, antagonists, inverse agonists, and so on) and inhibitors of interactions of nuclear receptors with various proteins related to the transcription. Here, we overview the current status of chemical screening of nuclear receptor modulators, covering the assay methods used for screening, available equipment, and chemical libraries for screening. We include some recent examples of the discovery of nuclear receptor modulators, especially those having ligands, while chemical screening is also useful to identify the specific modulators for orphan nuclear receptors, and we comment on prospects for the future.

## 2. Nuclear Receptors as Screening Targets

Nuclear receptor superfamily members are widely present in Metazoa, including humans, and other chordates, insects, and sponges. For example, the vitamin D receptor (VDR: NR1I1 as the logical number made by the International Committee of Pharmacology Committee on Receptor Nomenclature and Classification) has been cloned from lampreys, classified as an evolutionarily ancient vertebrate [18], and shown to bind with high affinity to 1α,25-dihydroxyvitamin D_3_, an active metabolite of vitamin D_3_ [19]. Indeed, 289 types of nuclear receptors have been reported in *Caenorhabditis elegans* [20] and 21 types in *Drosophila* [21]. Forty-eight types of nuclear receptors have been found in humans, and 25 of these are orphan receptors with unknown endogenous ligands [22]. The ligand-dependent action of nuclear receptors includes that the activated receptors translocate into the nucleus, bind to the specific sites of DNA, and regulate the target gene expression to elicit various events, such as cell proliferation, differentiation, reproduction, metabolism, and the maintenance of homeostasis [23,24]. Some orphan receptors are known to act constitutively as transcription-promoting factors and to play roles in releasing transcriptional repression [25]. The development of specific modulators of nuclear receptors is important not only to provide tools for fundamental functional studies, but also for research into human disease.

Nuclear receptors have evolved from a single gene and have been systematically classified based on their structural and functional domains, A/B–F (Figure 1) [26]. The A/B region contains a transcription-promoting region (activation function 1; AF-1). The C region located near the center of the receptor contains a zinc (Zn)-finger structure, and includes the DNA-binding domain (DBD) responsible for recognition and binding to a specific DNA sequence [27]. The D region, known as the hinge region, contains the nuclear localization signal (NLS). The E region includes the ligand-binding domain (LBD) and has a ligand-dependent transcriptional activation function (activation function 2; AF-2). The F region is an optional C-terminal domain. Steroid receptors such as the estrogen receptor (ER) (ERα: NR3A1, ERβ: NR3A2) and androgen receptor (AR: NR3C4) bind to DNA as homodimers. The retinoid X receptor (RXR) (RXRα: NR2B1, RXRβ: NR2B2, RXRγ: NR2B3) forms heterodimers with various receptors as partners to bind to their specific DNA sites. There are also some receptors that bind to DNA as monomers, including some orphan receptors [28,29]. The two AF regions regulate transcription by directly interacting with transcription cofactors, such as corepressors and coactivators [30,31,32].

The ligand-binding site is the primary target of many screenings, as ligand binding triggers transcriptional activity. Experimental and theoretical studies on ligand–receptor interactions and the conformational changes induced in the receptors provide a wealth of information for developing screening strategies and assay methods. Protein–protein interactions (PPIs) involving transcriptional co-regulators also play important roles in the regulation of nuclear receptor functions. Therefore, PPIs can also be targets for drug discovery, with both peptides [33] and chemical compounds [34,35] having the potential to disrupt these PPIs. Protein conformational changes and the activation mechanisms involved are described in detail in the reviews [28,36,37]. Other screening strategies may also be employed. As shown in Figure 1, nuclear receptors are regulated by various post-translational modifications, which are referenced in several well summarized reviews [35,38,39,40]. For example, it is possible to target processes that regulate the post-translational modifications of nuclear receptors, such as the ERα (Figure 1) [41,42]. For example, the phosphorylation of nuclear receptors regulates nuclear translocation and transcription, while ubiquitination plays a role in protein degradation. In addition, the binding of chaperones such as heat shock protein 90 (HSP90) and HSP70 controls the nuclear translocation of some receptors [43,44], so the chaperones can also be targets for screening strategies [45,46,47]. Some of the functions of nuclear receptors include the regulation of the methylation and acetylation of histones, both of which change the closed/open state of nucleosomes [48], and thus the regulatory mechanisms of these epigenetic controls can also serve as targets for the chemical screening of nuclear receptor functions [49,50].

## 3. Assay Methods Used for Screening

The ability to discover active compounds in a chemical library depends largely on the efficiency and robustness of the assay method used in the chemical screening, and also on the selection criteria for the chemical library. Assay methods used to evaluate nuclear receptors can be classified into two types: one is target-based chemical screening, which involves the direct observation of the interaction between the ligand and the target receptor, or the conformational changes of the target receptor, or the changes in transcriptional activity, and the other is phenotypic chemical screening [51,52].

### 3.1. Target-Based Chemical Screening

Target-based chemical screening involves the in vitro assay of a specific nuclear receptor or its components. The interaction with the target nuclear receptor can be directly investigated by using purified protein [53,54,55]. For example, one approach is to measure the fluorescence polarization of a probe that has been linked to a specific ligand [56,57,58]. Levison et al. first employed this sophisticated assay strategy for targeting nuclear receptors [59]. It is easier to design probes for assays that target receptors with known ligand-binding modes. Nuclear magnetic resonance (NMR) [60,61] and surface plasmon resonance (SPR) [62,63,64] are useful screening methods to investigate the direct interaction between compounds. These methods require minimal or no protein modification, but do require large amounts of purified protein, and offer relatively low throughput. Thus, these methods are more suitable for targeted secondary screening. One-hybrid or two-hybrid assay using yeast or animal cells is another example of target-based screening [65,66,67,68], and this strategy was first applied for assaying nuclear receptors by Kakidai et al. [69] and Webster et al. [70]. This approach can be used not only for studying the binding of a ligand molecule to a receptor, but also for studying the heterodimer formation of nuclear receptors and the transcriptional machinery. Robust assay systems have been constructed for elucidating the nature of biomolecular complexes. However, a limitation of this approach is that the activity of the reporter reflects the transcriptional activity of the nuclear receptors, and therefore the assays can select false-positive compounds that modulate the transcriptional activity via indirect mechanisms, including the activation of basic transcriptional machinery. The reporter assay using a specific DNA-binding sequence of a nuclear receptor is also a simple and useful target-based screening method [71,72,73].

### 3.2. Phenotypic Chemical Screening

Phenotypic chemical screening uses assays to detect the effects on biological phenomena that involve nuclear receptors (e.g., cell viability, alteration of cellular morphology) [17,24]. Cell proliferation assay is a very common screening method, and many kits and measuring instruments are available for its evaluation [12,13,74]. Morphological changes due to cell differentiation and changes in the secretion of physiologically active substances, cytokines, and enzymes resulting from differentiation are also useful targets for screening. Although the detection of these secreted molecules is possible by using a robust method such as the enzyme-linked immunosorbent assay, a limitation of this approach is that it involves a heterogeneous assay [75]. The development in the late 1990s of label-free devices, cellular dielectric spectroscopy, and automated cellular imaging and analysis instruments for cell morphological analyses has led to the widespread use of cell morphology-based screening [76,77,78,79]. 

In the case of phenotypic chemical screening, it is important to rescreen the hit compounds using another screening method in order to confirm whether the hits regulate the phenotype via the target nuclear receptor, even when cell lines with genetic modifications are used.

## 4. Equipment for Chemical Screening

Commercial equipment for chemical screening has been available since around the year 2000, although basic instruments such as plate readers and dispensers are adequate to achieve cost-effective and efficient screening. High-throughput screening-focused devices not only improve throughput by incorporating handling machines, such as automatic samplers, but also enable simultaneous measurements in multiplex assays [80,81]. Devices are now available to determine cell morphology and size, which have been difficult to measure with high throughput in the past, and also to measure extremely rapid cell responses such as calcium oscillations. An instrument that can perform cellomics analysis, called high-content analysis, has also greatly impacted screening assays [82,83]. It not only captures images by using automated bioimaging devices, but also performs image analysis, visualizes the data, and manages the images. For further details on high-throughput screening equipment, good review articles are available [84].

## 5. Assay Development and Planning of a Screening Scheme

An important aspect of preliminary screening is the speedy and reasonably reliable selection of active compounds from a large chemical library. Major considerations in the design of assay protocols are as follows:

Minimizing handling steps: the use of larger handling volumes and homogeneous assays (i.e., all assay components exist in a single phase (solution) at the time of detection) can improve reproducibility, especially for in vitro assays.

Variability while performing the assay: although all assays/experiments include some variability, minimizing variability is important for the selection of hit compounds and also to avoid selecting false-positive compounds [85,86]. 

Evaluation of the screening system: the capability and efficiency of the screening system depend on the suitability or quality of the assay used in the screening. The Z-factor is a useful tool for the comparison and evaluation of the quality of assays and can be utilized in assay optimization and validation. The Z’ factor is a measure of separation between maximum and minimum controls in an assay that takes account of the extent of variability in the assay [87]. It reflects both the dynamic range of the assay signal and the data variation associated with the signal measurements. A Z’ factor close to 1 is desirable, while 0.5 is acceptable. 

Even non-HTS assays can generate large amounts of data. For example, useful tools include small-scale chemical libraries such as the Library of Pharmacologically Active Compounds (LOPAC^®^1280), containing compounds with known functions that could facilitate the design of a screening system [88]. A general screening flowchart is shown in Figure 2. 

## 6. How to Select a Chemical library

### 6.1. General Aspects of Library Selection

Various chemical libraries are now available, including those of public organizations and pharmaceutical companies. Besides the number and diversity of the compounds in the library, the quality of the library is significant. Especially for drug discovery, various criteria are used for the selection of compounds to be included in libraries, such as long-term storage stability, the synthetic convenience of structural development, and drug-likeness as represented by the fulfillment of Lipinski’s rule of five [89]. Libraries can be classified by the type of the compounds they contain, as shown in Table 1.

### 6.2. Fragment and Scaffold Libraries

Fragment libraries can efficiently cover the chemical space with compounds of a molecular weight of less than 250, and provide information on the structure of the complex with the protein even in the early stages of target-based chemical screening. However, the low molecular weight range of the compounds may limit the applicable assay systems or targets, since the biological activity of compounds in fragment libraries is often weak. Fragment libraries can be employed with assay systems using X-ray crystal structure analysis, SPR, or NMR, although high concentrations (100–1000 µM) of the compounds still have to be used [90,91].

Scaffold libraries are composed of compounds with molecular weights of ≤350 (mainly 250–350) that could be based on specific molecular skeletons and allow the efficient coverage of the chemical space [92]. Compared to fragment libraries, the molecular weights of compounds in a scaffold library are slightly higher and the binding affinity tends to be higher. Therefore, it is possible to design biological assays with concentrations in the 10–200 µM range.

Hit compounds identified in preliminary screening with scaffold libraries can be modified to optimize activity. Both fragment and scaffold libraries explore target chemical spaces and enable the structural optimization of compounds. Thus, with the molecular weight being limited to ≤500 or to ≤160, there are still 10^60^ and 10^7^ types, respectively [93,94], of possible compound variations, which means that compound libraries containing 100,000 to several million fragment compounds can cover a vast chemical space [10]. As the molecular weight of the compounds in the library becomes larger, it becomes more difficult to cover the chemical space of a target such as a ligand-binding pocket. Moreover, if the molecular weight of a hit compound is large, this may impede subsequent optimization to enhance drug-likeness. It is important to note that fragment libraries have simple structures, and that computational science can greatly facilitate the analysis of the hits. The size of currently used fragment libraries ranges from several thousand to 20,000 [92,95,96] and such libraries are owned and utilized not only by pharmaceutical companies, but also by venture companies and academia.

Plexxikon identified hit compounds for all three-peroxisome proliferator-activated receptor (PPAR) subtypes: PPARα (NR1C1), PPARβ (NR1C2), and PPARγ (NR1C3) [97]. Based on X-ray structure analysis of the hit compounds, they found that the indole fragment has a unique binding mode, compared with other known PPAR ligands. Based on this discovery, structure-based drug design was performed to obtain the candidate compounds for treating type 2 diabetes mellitus. Although some candidates showed potential side effects, indeglitazar, developed as a full agonist of PPARα and a partial agonist of PPARγ, has progressed to phase II trials for Type 2 diabetes (T2D). 

### 6.3. Focused Library

A focused library is a compound library designed to have a high hit rate for a specific molecular species. There are various types of focused libraries, such as G-protein-coupled-receptor-focused libraries and kinase-focused libraries [98,99]. Generally, a focused library contains a rather small number of compounds, and is easy to handle, while hit compounds can be obtained with high probability. SCREEN-WELL^®^ offers the Nuclear Receptor Ligand Library that is focused on nuclear receptors [100]. ChemBridge provides a database of 5000 compounds for virtual screening and supplies selected compounds found to be capable of binding to nuclear receptors [101].

Tachibana et al. performed chemical screening for the activators of PPARα (NR1C1) using a two-step cell-based assay that regulates PPARα (NR1C1) expression via Tet [102]. They selected compounds that act directly on PPARα (NR1C1), rather than on the heterodimer with RXR (NR2B1-3) or various cofactors in the assays. Through chemical screening, Yang et al. found a chemical combination that enables the culture of human and mouse pluripotent stem cells (PSCs) bipotential towards both embryonic (Em) and extraembryonic (ExEm) cells [103]. They identified a chemical cocktail that gives both cell types chimeric abilities. Their idea was to use nuclear receptor ligands and protein kinase inhibitors that have a clear function, making efficient use of a target-focused compound library including these compounds. More than 100 primary hits were subjected to second screening to afford more than 30 small molecules as hit compounds. Although the final selected compounds were not a modulator of the nuclear receptor, various combinations of these small molecules were further tested to identify combinations that enable the long-term self-renewal of these colonies. Finally, they developed a cocktail of compounds that can support the regeneration of Em and ExEm lineages. Thus, when targeting effects that can only be achieved by combining two or more drugs, the focused library, which contains a limited number of compounds and has a clear point of action, is a powerful tool.

### 6.4. Natural Products Library

Because of the diversity of biological species, the physiological activities and chemical structures of natural compounds also show immense diversity, and many of them have proved useful as pharmaceuticals or leads for drug discovery [104]. To date, more than 200,000 natural compounds have been isolated from various biological resources, and their structures have been elucidated [105]. Although natural compounds are often isolated only in small amounts, recent progress in highly sensitive screening systems and devices means that the biological activities of even small amounts of compounds can now be detected. 

Although medical treatments have been established for prostate cancer, various resistance mechanisms still impede treatment. Using androgen receptor-positive LNCaP cells and negative PC-3 cells in an assay system, Xu et al. discovered 17β-hydroxywithanolides as anti-cancer candidate compounds from a natural products library [106]. Thus, homologous cancer cells with different nuclear receptor phenotypes are very useful for cell-based assays. Witanolide constitutes a class of steroidal lactones structurally based on the ergostane skeleton and is abundant in plants of the eggplant family. Although in vitro anti-cancer activity has already been demonstrated with some witanolides, this report was the first to reveal the inhibitory effect on TNF-α-induced NF-κB inhibition and its activity against prostate cancer. Tachibana et al. described a reporter assay system that can be used to assess PPARγ (NR1C3) and PPARδ (NR1C2) activities. This assay system was used to evaluate approximately 200 natural resource extracts and PPARγ (NR1C3) activators, and identified the alkaloid evodiamine from Evodia fruit as a hit [107]. In such assays, it is often necessary to purify and identify the active ingredients from plant extracts. 

### 6.5. Library of Function-Known Compounds

This type of library comprises compounds with known bioactivity, such as enzyme inhibitors and anti-cancer agents. Therefore, this category overlaps with focused libraries. It can be easier to identify the site of action when a hit compound has been obtained by the means of phenotypic chemical screening. In recent years, research efforts have focused on using over-the-counter drugs as therapeutic agents for diseases other than the original target diseases, i.e., so-called “drug repositioning” or “drug repurposing”. In particular, out-of-patent drugs can be used without any restrictions [101]. There is also a “drug repositioning compound library” that collects compounds from discontinued clinical or preclinical studies conducted by large pharmaceutical companies. As their pharmacological properties have already been confirmed, and their basic safety has been demonstrated, proof of activity in new drug efficacy evaluation systems can greatly shorten the duration of clinical trials compared with that in the case of conventional drug discovery focusing on novel compounds. Repositioned compounds can also be used as positive controls for HTS and for calibration when developing a new assay method.

Johnston et al. screened small molecules that inhibit the nuclear translocation of AR and identified three hits. They successfully set up a high-content screening (HCS) system, which detects the nuclear translocation of a protein fused with a nuclear receptor and green fluorescence protein (GFP). They used HCS to screen a library of more than 200,000 compounds at the National Institutes of Health (NIH) molecular library screening center network. The structures of the three hit compounds are different from those of known AR modulators, and the authors concluded they target cofactor proteins that the control nuclear translocation of AR. Thus, hit compounds obtained from phenotypic screenings are expected to become leads for new drug discovery targets [108]. Wiel et al. found 15 ligands that bind to the LBD of the farnesoid X receptor (FXR) (FXR: NR1H4, FXRβ: NR1H5), using a fluorescence resonance energy transfer (FRET) assay system that observes the conformational change of the FXR LDB in the presence of a peptide with LXXLL motifs [109]. The compounds clearly demonstrated agonistic activity, although they were less potent than a positive control, GW4064. Finally, it was found that compounds not similar to bile acids or GW4064 can bind to the FXR LBD. This screening used an FDA-approved compound library, and the hit compounds are already being used in clinical applications. 

### 6.6. Virtual Library/Virtual Screening

Since the 1990s, increasing amounts of data on protein structure and the development of protein-ligand docking simulations have enabled structure-based drug discovery (SBDD) [110]. In silico screening, also called virtual screening, is a computer simulation method to select compounds that are predicted to interact with a target protein. In addition to in-house libraries (tens of thousands to millions of compounds) owned by companies and research institutes, commercially available compounds are also used for in silico screening. Furthermore, in silico screening can cover the compounds never synthesized or isolated, which can be constructed according to the purpose of the screening. Thus, the likelihood of discovering active compounds has been rapidly increasing. 

Wang et al. identified a new RXRα tetramer stabilizer using structure-based virtual screening [111]. They obtained two hits from the FDA-approved small molecule drug registries in the DrugBank 4.0 collection. They showed that one of them, atorvastatin, binds to RXRα, and exerts an apoptotic effect through RXRα. Pang et al. discovered a selective estrogen receptor modulator by using a ligand-based machine learning method and structure-based molecular docking [112]. A total of 162 compounds were predicted as ER antagonists and were further evaluated by molecular docking. Although the top-ranked compounds in the docking were well known ER modulators, this study suggests that machine-learning methods will be useful for predicting the biological effects of candidate compounds.

There are more than 100 reports of screening processes for the modulators of nuclear receptors. ER, AR, and PPARs are major targets for chemical screening. The relative ease of obtaining cell lines, fusion proteins with fluorescent markers, and purified proteins for these assays and the ease of constructing assay systems are likely be among the reasons for the large number of screening studies. Typical examples using each type of compound library are summarized in Table 2, which provides important insights for developing a screening plan. 

## 7. How to Acquire a Library

In the United States, the National Institutes of Health (NIH) Roadmap for Medical Research calls for establishing a cooperative research network that will use large-scale screening methods to identify small molecules that can be used as biomedical research tools and in early drug discovery [154]. In the Roadmap, the NIH defines a number of areas that are crucial for the future of American medical research and emphasizes the need for these areas to be developed through a concerted national effort. The National Center for Advancing Translational Sciences (NCATS) Chemical Genomics Center (NCGC), formerly known as the NIH Chemical Genomics Center, is one of the centers of the Molecular Libraries Screening Centers Network within the NIH Roadmap for Molecular Libraries Initiative [155]. It has established networks of screening centers, technical development centers, compound libraries, and an online public database (called PubChem) of compound structures and the results of screenings [156]. In addition, the NIH established the Molecular Libraries Probe Production Centers Network in 2008 as a national resource center for innovative chemical tools that can be used in biomedical research [157]. The network is a nationwide consortium of small molecule screening centers, which also optimizes chemistry to produce chemical probes for targets or phenotypes that are to be examined with the help of assays. NCATS possesses chemical libraries that consist of over 590,000 functionally known and unknown compounds.

In Europe, the Chemical Genomics Centre is an initiative of the Max Planck Society in cooperation with European pharmaceutical companies that began in 2005 [158]. European ScreeningPort GmbH was founded in 2007 as a public–private partnership to accelerate the introduction of drug discovery services to the academic and biomedical research community [159,160]. The European Lead Factory (ELF) project seeks to address this challenge by leveraging the diverse knowledge and experience of academic groups as well as the small and medium enterprises (SMEs) engaged in synthetic and/or medicinal chemistry [161]. As part of the ELF open-source model, seven pharmaceutical companies (AstraZeneca, Bayer, Johnson Johnson, Lundbeck, Merck, Sanofi, and UCB) have contributed a total of 321,000 compounds from their proprietary collections [162]. 

In Asia, the mission of the Asian Chemistry Biology Initiative is to speed up Asian chemical biology research by promoting international collaborations and by sharing research resources among Japan, Korea, China, Singapore, the United Arab Emirates, and New Zealand [163]. Furthermore, the mission aims to promote chemical biology in emerging Asian countries (Vietnam, the Philippines, and Thailand) by recruiting and training the brightest graduate students from those countries [164,165]. The Chinese National Compound Library (CNCL), located in Shanghai Zhangjiang Hi-tech Park (the “Pharma Valley” of China), is a major research and development establishment managed by the National Center for Drug Screening; Shanghai Institute of Materia Medica; Chinese Academy of Sciences; and Shanghai Zhangjiang Biopharmaceutical Base Development, Co., Ltd. As of 2015, its storage capacity has approached two million diversified compounds. Along with this library, advanced sample handling, information management, and quality control systems will be included. As a valuable source of material and information, the CNCL will collaborate with both domestic and international stakeholders to promote the sustained development of the Chinese pharmaceutical industry [166].

In Japan, the Japanese Society for Chemical Biology was founded in 2005 [167,168], and its first annual meeting was held in 2006. For the purpose of making use of the intellectual property of universities in industry and to contribute to studies in the life sciences, the Drug Discovery Initiative, DDI, was founded at the University of Tokyo in 2006 [169]. A high-quality chemical library is important for developing synthetic small molecules that regulate biological functions. Therefore, the DDI has collected more than 210,000 compounds and supported the utilization and application of this chemical library. The DDI’s chemical library is freely available, but users must report their experimental results. The name “Chemical Biology Research Initiative” was changed to “Open Innovation Center for Drug Discovery” in 2011. Furthermore, in 2012, the BINDS program (Basis for Supporting Innovative Drug Discovery and Life Science Research) was created to promote the development of innovative processes for drug design and medical technology [170]. To achieve this goal, the Basis for Supporting Innovative Drug Discovery and Life Science Research is accelerating the collection of results and technical bases utilizable for the drug design process, as well as the libraries of publicly available chemical compounds. The Chemical Biology Core facility at RIKEN has constructed chemical libraries, a Natural Products Depository called NPDepo, and databases by using the results of genetic and organic chemical research on secondary metabolites from microorganisms and natural products [171]. NPDepo also promotes basic research to validate the usefulness of their chemical libraries, to identify the molecular targets of active compounds, and to elucidate the molecular mechanisms of action of the active compounds in their libraries [172]. 

## 8. Conclusions and Perspectives

Many chemical screening studies targeting nuclear receptors for drug discovery have been reported, and many physiologically or pharmacologically active substances have been identified. Nuclear receptors are controlled not only by specific ligands, but also by multiple regulatory mechanisms involving receptor modifications such as epigenetic, ubiquitination, and protein maturation, as well as other presently unknown mechanisms. The screening of specific modulators for nuclear receptors will provide essential tools for developing a detailed understanding of the regulation of nuclear receptors, and for clinical applications. We hope that the information provided in this review will assist researchers in the development of screening procedures targeting nuclear receptor modulators.

## Figures and Tables

**Figure 1 ijms-21-05512-f001:**
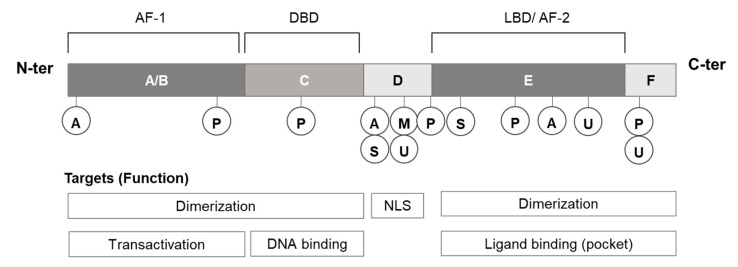
Summary of the functional domains of nuclear receptors, and the post-translational modification sites in ERα: NR3A1; A: acetylation, M: methylation, P: phosphorylation, S: SUMOylation, U: ubiquitination.

**Figure 2 ijms-21-05512-f002:**
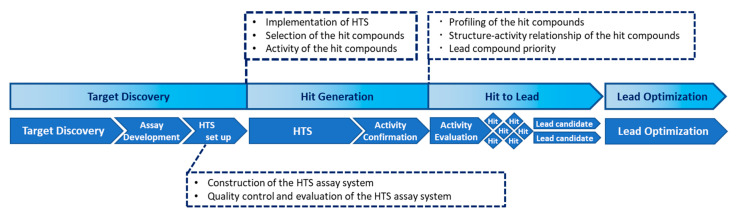
Chemical biology workflow: the role of each step and implementation requirements.

**Table 1 ijms-21-05512-t001:** Library features and assay applicability.

Library	Number of Compounds	Molecular Weight	Appropriate Assay	Typical Concentration Range of Compounds Examined
Fragment	100–10,000	≤250	X-ray crystal structure analysis, NMR, Surface plasmon resonance, Fluorescence correlation spectroscopy	100–1000 µM
Scaffold	100–10,000	250–350	Cell proliferation assay, Reporter gene assay, Phenotype assay	Tens of µM-200 µM
Focused	Depends on owning institution	Depend on target proteins	Cell proliferation assay, Reporter gene assay, Phenotype assay	Depends on assay
Natural product	Depends on owning institution	500–2000	Cell proliferation assay, Reporter gene assay, Phenotype assay	Depends on assay
Virtual	Millions or more	100–500	In silico screening	Not applicable

**Table 2 ijms-21-05512-t002:** Representative examples of the screening of nuclear receptors.

Library	Target Nuclear Receptor ^a^
General	ER [113,114,115,116,117,118], AR [108,119], PR [120], RAR [121], TR [122,123], PPAR [102,124,125]
Fragment and Scaffold	ER [116,126], AR [127,128], PR [129,130], RAR [131], VDR [132], TR [133], PPAR [118,119], LXR [134], PXR [135]
Natural	ER [136,137], AR [120], PR [120,138], RAR [139], VDR [140], TR [123]
Function known	ER [116,141], AR [138], GR [142], FXR [58], CAR [143], PXR [111,144]
Virtual	ER [145,146,147], AR [148,149,150,151,152,153], RXR [111]

^a^ Numbers in parentheses are the reference numbers. See the list of abbreviations about the logical number of each nuclear receptor made by the International Committee of Pharmacology Committee on Receptor Nomenclature and Classification.

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
