# Peer review of "Chemical Screening of Nuclear Receptor Modulators"

_ijms, 2020, doi:10.3390/ijms21155512_

Round 1

Reviewer 1 Report

This review entitled “Chemical screening of nuclear receptor modulators” by Ishigami-Yuasa and Kagechika is focused on assays methods and chemical libraries for screening.

The text of this review is scientifically accurate and well documented, and attempts to address interesting new issues that have emerged in recent years regarding chemical libraries for screening of nuclear receptor modulators. Overall this is an interesting and timely review that does bear significant and clear information and will be appropriate for a general audience. I think the manuscript is suitable for publication if the text is improved.

The title and abstract are informative and give a clear idea of what to expect from the paper. However, the authors should specify that this review is mainly concerned with nuclear receptors having a ligand.

The second part of the review on the chemical libraries is sufficiently detailed and clear while the parts relating to nuclear receptors and the assay methods for chemical screening are more superficial. It is likely not easily comprehensible for a reader which poorly knows about it.  But the cited literature is abundant and recent which allows the reader with a certain effort to obtain more information and details.

In the introduction section (I) the authors refer to different types of nuclear receptor modulators. It would be advisable to specify what these different types of compounds and their functionalities are and to define them.

The second part (Nuclear receptors as screening targets) needs to be less confusing regarding the different classes of nuclear receptors (orphans vs. receptors with ligands). The general mechanism of action proposed for NR applies only to a limited number of them (bottom of page 2). This part seems to me to be insufficiently detailed.

Authors use trivial names for nuclear receptors. A logical numbering system and receptor code, supporting the trivial names, was made by the International Committee of Pharmacology Committee on Receptor Nomenclature and Classification (NC-IUPHAR). In each manuscript dealing with nuclear receptors, it is recommended that the receptors be identified by the official names at least once in the summary and the introduction. Once the name has been established, authors may use the trivial name for the remainder of the manuscript.  For instance, the trivial name and the formal nomenclature for VDR is NR1I1; VDR being the trivial name.

Author Response

Response to the comments by reviewer #1

We revised the manuscript according to the reviewers’ comments as follows. The revised parts are shown in red color in the text of the manuscript file.

Comments from Reviewer #1

  1. Reviewer’s comment: The title and abstract are informative and give a clear idea of what to expect from the paper. However, the authors should specify that this review is mainly concerned with nuclear receptors having a ligand. 

Our response: In this review, we mainly treat the chemical screening for nuclear receptors having a ligand, while it is also useful to identify the specific modulators for orphan nuclear receptors. We revised the sentences in the section of Introduction as follows (revised parts are shown by underline).

(Before revision)Here, we overview the current status of chemical screening of nuclear receptor modulators, covering the assay methods used for screening, available equipment, and chemical libraries, and we comment on prospects for the future.

(After revision) Here, we overview the current status of chemical screening of nuclear receptor modulators, covering the assay methods used for screening, available equipment, and chemical libraries for screening. We include some recent examples of the discovery of nuclear receptor modulators, especially those having ligands, while chemical screening is also useful to identify the specific modulators for orphan nuclear receptors, and we comment on prospects for the future.

  1. Reviewer’s comment: The second part of the review on the chemical libraries is sufficiently detailed and clear while the parts relating to nuclear receptors and the assay methods for chemical screening are more superficial. It is likely not easily comprehensible for a reader which poorly knows about it. But the cited literature is abundant and recent which allows the reader with a certain effort to obtain more information and details.

Our response: According to the reviewer’s comments, we revised the sentences in the parts relating to nuclear receptors about the modulation of nuclear receptor actions with recent papers. As for the section of assay methods for chemical screening, the examples are shown in the section of chemical libraries, besides the cited references, which will be helpful for the readers to understand the each method. The revised sentences are as follows (revised parts are shown in underline).

(Before revision) Therefore, PPIs can also be targets for drug discovery, with both peptides [32] and chemical compounds [33, 34] having the potential to disrupt these PPIs. Other screening strategies could be adopted, such as targeting the processes controlling post-translational modification of nuclear receptors, as in the case of ERα(Fig. 1) [35, 36].

(After revision) Therefore, PPIs can also be targets for drug discovery, with both peptides [33] and chemical compounds [34, 35] having the potential to disrupt these PPIs. Protein conformational changes and the activation mechanisms involved are described in detail in the reviews [28, 36, 37]. Other screening strategies may also be employed. As shown in Fig. 1, nuclear receptors are regulated by various post-translational modifications, which are referenced in several well-summarized reviews [35, 38-40]. For example, it is possible to target processes that regulate the post-translational modifications of nuclear receptors, such as the ERα (Fig. 1) [41, 42].

(Added references)

  1. Weikum, E.R.; Liu, X.; Ortlund, E.A. The nuclear receptor superfamily: A structural perspective. Protein Sci. 2018, 27, 1876-1892.
  2. Kumar, R.; McEwan, I.J. Allosteric modulators of steroid hormone receptors: structural dynamics and gene regulation. Rev. 2012, 33, 271-299.
  3. Cromm, P.M.; Crews, C.M. Targeted Protein Degradation: from Chemical Biology to Drug Discovery. Cell Chem. Biol. 2017, 24, 1181-1190.
  4. El Hokayem, J.; Amadei, C.; Obeid, J.P.; Nawaz, Z. Ubiquitination of nuclear receptors. Sci. (Lond). 2017, 131,917-934.
  5. Helzer, K.T.; Hooper, C.; Miyamoto, S.; Alarid, E.T. Ubiquitylation of nuclear receptors: new linkages and therapeutic implications. Mol. Endocrinol. 2015, 54, R151-R167.

  1. Reviewer’s comment: In the introduction section (I) the authors refer to different types of nuclear receptor modulators. It would be advisable to specify what these different types of compounds and their functionalities are and to define them.

Our response: According to the reviewer’s comments, we added the following sentences about the nuclear receptor modulators in the text (page 2, the last paragraph of Introduction section, line 4).

(Added sentences) In this review, we focus on nuclear receptors as target biosubstances. Nuclear receptors are ligand-inducible transcription factors [17]. There are several types of modulators of nuclear receptors, including the specific ligands (agonists, antagonists, inverse agonists, and so on) and inhibitors of interactions of nuclear receptors with various proteins related to the transcription.

  1. Reviewer’s comment: The second part (Nuclear receptors as screening targets) needs to be less confusing regarding the different classes of nuclear receptors (orphans vs. receptors with ligands). The general mechanism of action proposed for NR applies only to a limited number of them (bottom of page 2). This part seems to me to be insufficiently detailed.

Our response: According to the reviewer’s comments, we revised the sentences as follows (revised parts are shown by underline).

(Before revision) Nuclear receptors bind to ligands, translocate into the nucleus, and act there as transcription factors to regulate various events, such as cell proliferation, differentiation, reproduction, metabolism, and the maintenance of homeostasis [22, 23]. Therefore, development of specific modulators of nuclear receptors is important not only to provide tools for fundamental functional studies, but also for research into human disease.

(after revision) Ligand-dependent action of nuclear receptors includes that the activated receptors translocate into the nucleus, binds to the specific sites of DNA, and regulate the target gene expression to elicit various events, such as cell proliferation, differentiation, reproduction, metabolism, and the maintenance of homeostasis [23, 24]. Some orphan receptors are known to act constitutively as transcription-promoting factors and to play roles in releasing transcriptional repression [25]. Development of specific modulators of nuclear receptors is important not only to provide tools for fundamental functional studies, but also for research into human disease.

  1. Reviewer’s comment: Authors use trivial names for nuclear receptors. A logical numbering system and receptor code, supporting the trivial names, was made by the International Committee of Pharmacology Committee on Receptor Nomenclature and Classification (NC-IUPHAR). In each manuscript dealing with nuclear receptors, it is recommended that the receptors be identified by the official names at least once in the summary and the introduction. Once the name has been established, authors may use the trivial name for the remainder of the manuscript. For instance, the trivial name and the formal nomenclature for VDR is NR1I1; VDR being the trivial name.

Our response: Thank you for your suggestion. We added the logical numbering with trivial name of each nuclear receptor in the text including that in list of Abbreviations. In Table 2, addition of the logical number complicate the table, and therefore I just added the following sentence in the legend as follows (shown by underline).

(Revised legend of Table 2) Superscripts are the reference numbers. See the list of Abbreviations about the logical number of each nuclear receptor made by the International Committee of Pharmacology Committee on Receptor Nomenclature and Classification.

Reviewer 2 Report

In this review Ishigami-Yuasa and Kagechika discuss advances in small molecule screening for modulators of nuclear receptors. Nuclear receptors are a large family of mostly ligand-activated transcription factors and are important drug targets in metabolic disorders, inflammation, reproductive health and cancer. The review considers different screening strategies and the types of small molecule libraries that are available. The review is timely and well written and represents a valuable addition to the field.

There are however some points the authors should consider:

  1. Figure 1 nicely summarises the domain organisation of nuclear receptors and highlights the many post-translational modifications these proteins are substrates for. However, it would be good to briefly expand on this topic in the text and cite relevant literature or suitable review to allow the reader to find out more information.
  2. On page 8 there is a discussion of using pluripotent stem cells but the connection to nuclear receptors or small molecule screening is not clear? Were these cells used to identify modulators of particular nuclear receptors- a little more detail here would be good.
  3. On the same page the authors discus natural product libraries and cite a paper by Xu et al (101). However this paper appears to be about kidney cell cancer and not prostate cancer as discussed in the text. Please clarify.
  4. Table 2 is a nice summary of use of different libraries/ approaches to identify modulators of a number of nuclear receptors. It’s not clear if this is just meant to just show representative examples or to be more comprehensive. For example, the Rennie group in Vancouver have used virtual screening to identify modulators of the androgen receptor DBA-binding domain.  If it is only meant to show some examples this should be made clear.

Author Response

Response to the comments by reviewer #2

We revised the manuscript according to the reviewers’ comments as follows. The revised parts are shown in red color in the text of the manuscript file.

Comments from Reviewer #2

  1. Reviewer’s comment: Figure 1 nicely summarises the domain organisation of nuclear receptors and highlights the many post-translational modifications these proteins are substrates for. However, it would be good to briefly expand on this topic in the text and cite relevant literature or suitable review to allow the reader to find out more information.

Our response: Thank you for your suggestion. We added the following references according to the suggestion.

  1. Weikum, E.R.; Liu, X.; Ortlund, E.A. The nuclear receptor superfamily: A structural perspective. Protein Sci. 2018, 27, 1876-1892.
  2. Kumar, R.; McEwan, I.J. Allosteric modulators of steroid hormone receptors: structural dynamics and gene regulation. Rev. 2012, 33, 271-299.
  3. Cromm, P.M.; Crews, C.M. Targeted Protein Degradation: from Chemical Biology to Drug Discovery. Cell Chem. Biol. 2017, 24, 1181-1190.
  4. El Hokayem, J.; Amadei, C.; Obeid, J.P.; Nawaz, Z. Ubiquitination of nuclear receptors. Sci. (Lond). 2017, 131,917-934.
  5. Helzer, K.T.; Hooper, C.; Miyamoto, S.; Alarid, E.T. Ubiquitylation of nuclear receptors: new linkages and therapeutic implications. Mol. Endocrinol. 2015, 54, R151-R167.

  1. Reviewer’s comment: On page 8 there is a discussion of using pluripotent stem cells but the connection to nuclear receptors or small molecule screening is not clear? Were these cells used to identify modulators of particular nuclear receptors- a little more detail here would be good.

Our response: Yang et al. screened the drugs from focused libraries containing nuclear receptor modulator because nuclear receptors are important variables for development and regeneration. This idea is important and their experiment is significant example of chemical screening, although the hit compounds are not nuclear receptor ligands. We revised the sentences as follows (revision parts are shown by underline).

(Before revision) Their idea was to use nuclear receptor ligands and protein kinase inhibitors that have a clear function, making efficient use of a target-focused compound. More than 100 primary hits were subjected to second screening to afford more than 30 small molecules as hit compounds. Various combinations of these small molecules were further tested to identify combinations that enable long-term self-renewal of these colonies.

(After revision) Their idea was to use nuclear receptor ligands and protein kinase inhibitors that have a clear function, making efficient use of a target-focused compound library including these compounds. More than 100 primary hits were subjected to second screening to afford more than 30 small molecules as hit compounds. Although the final selected compounds were not a modulator of the nuclear receptor, various combinations of these small molecules were further tested to identify combinations that enable long-term self-renewal of these colonies.

  1. Reviewer’s comment: On the same page the authors discus natural product libraries and cite a paper by Xu et al (101). However this paper appears to be about kidney cell cancer and not prostate cancer as discussed in the text. Please clarify.

Our response: Thank you for pointing it out. This was an error in the citation and it has been corrected. We corrected the reference 107 (former reference number 101).

  1. Reviewer’s comment: Table 2 is a nice summary of use of different libraries/ approaches to identify modulators of a number of nuclear receptors. It’s not clear if this is just meant to just show representative examples or to be more comprehensive. For example, the Rennie group in Vancouver have used virtual screening to identify modulators of the androgen receptor DBA-binding domain. If it is only meant to show some examples this should be made clear.

Our response: We thank the reviewer for this pertinent comment. We have changed the Table 2 and added the following references.

  • Thank you for your suggestion. We added references as follows: Ref.150-158 (Table 2) Bafna, D.; Ban, F.; Rennie, P.S.; Singh, K.; Cherkasov, A. Computer-Aided Ligand Discovery for Estrogen Receptor Alpha. J. Mol. Sci. 2020, 21, 4193.
  • Singh, K.; Munuganti, R.S.N.; Lallous, N.; Dalal, K.; Yoon, J.S.; Sharma, A.; Yamazaki, T.; Cherkasov, A.; Rennie, P.S. Benzothiophenone Derivatives Targeting Mutant Forms of Estrogen Receptor-α in Hormone-Resistant Breast Cancers. J. Mol. Sci. 2018, 19, 579.
  • Singh, K.; Munuganti, R.S.N.; Leblanc, E.; Lin, Y.L.; Leung, E.; Lallous, N.; Butler, M.; Cherkasov, A.; Rennie, P.S. In silico discovery and validation of potent small-molecule inhibitors targeting the activation function 2 site of human oestrogen receptor α. Breast Cancer Res. 2015, 17, 27.
  • Carabet, L.A.; Lallous, N.; Leblanc, E.; Ban, F.; Morin, H.; Lawn, S.; Ghaidi, F.; Lee, J.; Mills, I.G.; Gleave, M.E.; Rennie, P.S.; Cherkasov. A. Computer-aided drug discovery of Myc-Max inhibitors as potential therapeutics for prostate cancer. J. Med. Chem. 2018, 160, 108-119.
  • Dalal, K.; Ban, F.; Li, H.; Morin, H.; Roshan-Moniri, M.; Tam, K.J.; Shepherd, A.; Sharma, A.; Peacock, J.; Carlson, M.L.; LeBlanc, E.; Perez, C.; Duong, F.; Ong, C.J.; Rennie, P.S.; Cherkasov, A. Selectively targeting the dimerization interface of human androgen receptor with small-molecules to treat castration-resistant prostate cancer. Cancer Lett. 2018, 437, 35-43.
  • Dalal, K.; Munuganti, R.; Morin, H.; Lallous, N.; Rennie, P.S.; Cherkasov, A. Drug-Discovery Pipeline for Novel Inhibitors of the Androgen Receptor. Methods Mol. Biol. 2016, 1443, 31-54.
  • Li, H.; Ban, F.; Dalal, K.; Leblanc, E.; Frewin, K.; Ma, D.; Adomat, H.; Rennie, P.S.; Cherkasov, A. Discovery of small-molecule inhibitors selectively targeting the DNA-binding domain of the human androgen receptor. Med. Chem. 2014, 57, 6458-6467.
  • Lack, N.A.; Axerio-Cilies, P.; Tavassoli, P.; Han, F.Q.; Chan, K.H.; Feau, C.; LeBlanc, E.; Guns. E.T.; Guy, R.K.; Rennie, P.S.; Cherkasov, A. Targeting the binding function 3 (BF3) site of the human androgen receptor through virtual screening. Med. Chem. 2011, 54, 8563-8573.
  • Axerio-Cilies, P.; Lack, N.A.; Nayana, M.R.; Chan, K.H.; Yeung, A.; Leblanc, E.; Guns, E.S.; Rennie, P.S.; Cherkasov, A. Inhibitors of androgen receptor activation function-2 (AF2) site identified through virtual J. Med. Chem. 2011, 54, 6197-6205.